# Dogs Can Be Reservoirs of *Escherichia coli* Strains Causing Urinary Tract Infection in Human Household Contacts

**DOI:** 10.3390/antibiotics12081269

**Published:** 2023-08-01

**Authors:** Peter Damborg, Mattia Pirolo, Laura Schøn Poulsen, Niels Frimodt-Møller, Luca Guardabassi

**Affiliations:** 1Department of Veterinary and Animal Sciences, University of Copenhagen, 1870 Frederiksberg, Denmark; pedam@sund.ku.dk (P.D.); mapi@sund.ku.dk (M.P.); laurajo.tw@gmail.com (L.S.P.); 2Department of Clinical Microbiology, Hvidovre Hospital, 2650 Hvidovre, Denmark; niels.frimodt-moeller@regionh.dk

**Keywords:** *Escherichia coli*, transmission, whole-genome sequence, One Health

## Abstract

This study aimed to investigate the role played by pets as reservoirs of *Escherichia coli* strains causing human urinary tract infections (UTIs) in household contacts. Among 119 patients with community-acquired *E. coli* UTIs, we recruited 19 patients who lived with a dog or a cat. Fecal swabs from the household pet(s) were screened by antimicrobial selective culture to detect *E. coli* displaying the resistance profile of the human strain causing UTI. Two dogs shed *E. coli* isolates indistinguishable from the UTI strain by pulsed-field gel electrophoresis. Ten months later, new feces from these dogs and their owners were screened selectively and quantitatively for the presence of the UTI strain, followed by core-genome phylogenetic analysis of all isolates. In one pair, the resistance phenotype of the UTI strain occurred more frequently in human (10^8^ CFU/g) than in canine feces (10^4^ CFU/g), and human fecal isolates were more similar (2–7 SNPs) to the UTI strain than canine isolates (83–86 SNPs). In the other pair, isolates genetically related to the UTI strain (23–40 SNPs) were only detected in canine feces (10^5^ CFU/g). These results show that dogs can be long-term carriers of *E. coli* strains causing UTIs in human household contacts.

## 1. Introduction

Approximately 80% of community-acquired urinary tract infections (UTIs) are caused by *Escherichia coli* [1]. Most strains causing UTIs are extraintestinal pathogenic *E. coli* (ExPEC) [2], which is a pathotype characterized by the presence of specific virulence genes such as adhesins, toxins, and polysaccharide coatings enhancing pathogenicity outside the intestinal tract [3]. In addition to humans, ExPEC-like strains of *E. coli* have been identified in various animal species, particularly dogs [4,5,6].

Although UTI patients are usually infected with strains colonizing their intestinal tract [7], several authors have emphasized the risk that pathogenic *E. coli* may be acquired via food of animal origin [8] or by contact with household pets [4]. It has been shown that healthy pets (mainly dogs) and humans living together frequently shed the same *E. coli* strain [9,10], and that household pets can be colonized with the strain causing UTIs in the owner [10,11,12,13]. Furthermore, exposure to dogs has been identified as a risk factor for development of multidrug-resistant *E. coli* UTIs in women [14]. On the other hand, *E. coli* is also the most common pathogen in canine UTIs and other common extra-intestinal infections in dogs such as pyometra. Molecular epidemiological studies from various countries have shown that the prevalent lineages among canine clinical isolates include dog-associated sequence type (ST) 372 as well as lineages that have been traditionally associated with humans (e.g., ST12, ST73, ST127, and ST141) [15,16]. Altogether, these data suggest that some UTIs may be zoonoses in either direction (human-to-pet or pet-to-human). However, the frequency of such transmission remains unknown.

The aim of this study was to investigate the role played by pets as reservoirs of *E. coli* strains causing UTIs in their human household contacts. For this purpose, we contacted pet household contacts who suffered from community-acquired *E. coli* UTIs and screened their pets selectively for occurrence of the UTI-causing strain. When a pet was found to carry a strain indistinguishable from that causing infection in its household contact, quantitative shedding of the UTI-causing strain by the pet and the owner was followed up after 10 months, and whole-genome sequencing (WGS) of pet and human isolates was used to infer their genetic relatedness.

## 2. Results

Among 119 eligible UTI patients, 19 (16%) were pet owners and agreed to participate in the study. Since 2 of the patients had 2 pets each, in total 21 pets were sampled, including 6 cats and 15 dogs. Antimicrobial selective culture indicated potential human–pet strain sharing for 7 of the 19 patients. Canine isolates from two pairs (A and B) displayed the exact same antibiotic susceptibility pattern of the UTI-causing strain and were indistinguishable from the strain by PFGE analysis (Table 1, Appendix A). Pair A comprised a 69-year-old woman and her dog, who had lived together for 3 years. Pair B consisted of a 53-year-old woman and her dog, who had been living together for 9 years. A third dog (pair C) shed *E. coli* harboring the same extended-spectrum beta-lactamase (ESBL) gene (*bla*_CTX-M-15_) found in the human UTI strain but displayed unrelated antimicrobial susceptibility and PFGE patterns (Table 1, Appendix A). Furthermore, *bla*_CTX-M-15_ was located on different plasmids in the pet isolate (104 kb plasmid that was non-typeable by PBRT) and in the UTI-causing strain (138 kb *IncF* plasmid). The remaining canine isolates were not genetically related to the human UTI strain isolated from their household contact by PFGE analysis (Appendix A).

Ten months later, bacterial counts of dog and human feces from pairs A and B were analyzed on MacConkey agar supplemented with the antibiotics to which the UTI-causing strain was resistant (ampicillin in combination with trimethoprim for pair A and ampicillin in combination with sulfadiazine for pair B). In pair A, coliforms displaying the ampicillin/trimethoprim resistance profile were more abundant in the feces of the patient (2.6 × 10^8^ CFU/g, 100% of total coliforms) than in the feces of the dog (4.3 × 10^4^ CFU/g, 0.01% of total coliforms). In pair B, coliforms displaying the ampicillin/sulfadiazine resistance profile of the UTI-causing strain were only detected in the feces of the dog (4.4 × 10^5^ CFU/g, 4.6% of total coliforms).

Upon WGS, core-genome alignment was performed on 14 isolates from pair A, including the UTI strain, 6 fecal isolates from the patient, and 7 fecal isolates from the dog, and on 8 isolates from pair B, including the UTI strain and 7 fecal isolates from the dog. A phylogenetic tree based on the alignment is presented in Figure 1A (pair A) and Figure 1B (pair B). There was a maximum of 88 and 61 SNPs between any two isolates from within pairs A and B, respectively (Appendix A). The UTI strain differed from the initial dog fecal isolate by only 2 SNPs in pair A and 17 SNPs in pair B. In pair A, the UTI-causing strain differed from the follow-up fecal isolates from the patient and the dog by 2–7 and 83–86 SNPs, respectively (Appendix A). In pair B, the UTI-causing strain differed from the follow-up canine isolates by 23–40 SNPs (Appendix A).

All human and canine isolates from pair A belonged to ST998 (phylogroup B2), harbored 3 antibiotic resistance genes (*ant(3″)-Ia*, *bla*_TEM-1A_ and *dfrA1*) on a Tn*7* transposon located on the chromosome (Figure 2A), and harbored 35 predicted virulence genes, of which 6 were associated with extra-intestinal pathogenic *E. coli* (ExPEC; *papA*, *papC*, *sfaDSE* and *kpsM* II; Table 2). All isolates from pair B belonged to ST80 (phylogroup B2) and harbored *bla*_TEM-1B_, *aph(3″)-Ib*, *aph(6)-Id* and *sul2* on an 11 kb ColRNAI plasmid (named pPD) (Figure 2B) and 38 predicted virulence genes, of which 2 were associated with ExPEC (*sfaD* and *kpsM* II; Table 2).

## 3. Discussion

We investigated possible transmission of *E. coli* strains shared by UTI patients and their pets using a longitudinal study design that combined selective screening for the UTI-causing strain in the feces of the two hosts and SNP analysis of human and pet isolates.

The UTI strain was detected within two weeks of the UTI incident in the feces of the household pet for 2 of the 19 (11%) patients enrolled in the study, as demonstrated by PFGE analysis. This result was supported by WGS analysis showing only 2 and 17 SNPs between the human UTI strain and the first canine isolate in pairs A and B, respectively. Although there is no universal consensus on the interpretation of SNP differences, a 17 SNP threshold was recently defined to infer *E. coli* transmission among hospital patients [17]. The arbitrary application of this threshold therefore supports direct or indirect strain interchange between UTI patients and their dogs in these two households.

A follow-up study conducted 10 months later to further explore the source of the UTI strains in the two households revealed two different scenarios. In pair A, both the dog and the household contact shed the *E. coli* ST988 strain displaying the same resistance phenotype as the original UTI isolate, but the fecal counts were much higher in the household contact. Furthermore, the 6 human isolates differed by only 2–7 SNPs from the original UTI isolate, whereas the canine follow-up isolates differed by 83–86 SNPs. Considering the estimated mutation rate of *E. coli*, ranging from 2.3 × 10^−7^ to 3.0 × 10^−6^ substitutions per site per year [18,19], along with the average genome size of *E. coli* (5.2 Mbp), the observed number of SNPs between the original and follow-up human isolates of pair A falls within the expected range of SNPs (1–13) that would arise in 10 months. Taken together, these results indicate that the household contact was a persistent fecal shedder of the UTI strain and most likely her own source of infection. In pair B, the UTI-causing ST80 strain was only detected in dog feces and displayed 23–40 SNP differences compared to the strain isolated 10 months before from urine of the household contact. Fairly short genetic distances (down to 13 SNPs, Appendix A) were shown between the first canine isolate and those obtained 10 months later, suggesting that this dog was a long-term carrier of the strain that caused infection in the owner 10 months before. According to the information we received on the dog–human interaction of pair B, the dog resided regularly in the household contact’s bed and sofa, and it was periodically fed with the same food consumed by its household contact. Together with the genetic strain analyses, this supports the possibility for a dog-to-human transmission event in pair B, but other possibilities, such as transmission via a common food source, cannot be ruled out.

The identified resistance genes correlated 100% with the observed phenotypes (Table 1 and Table 2). All isolates from pair A had genes conferring resistance to ampicillin (*bla*_TEM-1A_) and trimethoprim (*dfrA1*), whereas all isolates from pair B had genes encoding resistance to ampicillin (*bla*_TEM-1B_) and sulfadiazine (*sul2*). Genes conferring resistance to spectinomycin (*ant(3″)-Ia*, *aph(3″)-Ib, aph(6)-Id*) and streptomycin (*ant(3″)-Ia*) were additionally found by WGS, but these two antimicrobials were not included in the antimicrobial panel of the MIC test. Human and canine isolates shared the same virulence markers, including some associated with ExPEC; the group 2 capsular polysaccharide unit (*kpsM* II); S fimbriae subunits (*sfa* genes); and—in pair A—P fimbriae genes (*papA* and *papC*) [20]. Furthermore, isolates from both pairs were classified into phylogroup B2, which has historically been recognized as the most common group linked to ExPEC infections in humans [21]. The STs assigned to the sequenced isolates, specifically ST998 in pair A and ST80 in pair B, are not among the major global ExPEC lineages, but both have previously been associated with ExPEC strains in both dogs and humans [16,22,23,24,25].

A study by Johnson et al. [12] has documented the sharing of *E. coli* ST131 with *bla*_CTX-M-15_ by a pet and a child living in the same family household. We detected the same ESBL gene in canine and human isolates from pair C (Table 1), but the isolates were genetically unrelated, and *bla*_CTX-M-15_ was located on different plasmids. This suggests that the dog was not directly implicated in the UTI of the owner but does not exclude that *bla*_CTX-M-15_ might have transferred between the two hosts prior to insertion into different plasmids. Frequent insertion of *bla*_CTX-M-15_ linked to ISEcp1 has been hypothesized based on sequence analysis of CTX-M-15 plasmids in clinical *E. coli* of human and animal origin [26].

The main limitation of the study is the small sample size. It would have been desirable to enroll more than 19 pet–patient pairs, especially with cats, but the prevalence of pet owners agreeing to participate in the study was relatively low (approximately 15%) among the 119 UTI patients that were contacted by phone. Nevertheless, to our knowledge, this study is the first to provide an estimate on the extent of UTI strain sharing between pets and humans. Similar to the small sample size, it can be argued that a higher frequency of sampling pets and household contacts in the follow-up longitudinal study would have provided a more dynamic picture of strain shedding and diversity over time. However, even analyzing a higher number of samples would not have led to any firm conclusion concerning strain origin and direction of transmission.

## 4. Materials and Methods

### 4.1. Patient Recruitment and Sampling of Pets

The protocol for patient recruitment was approved by the National Committee on Health Research Ethics (journal record: H-4-2013-FSP-071). From February to May 2014, patients diagnosed at the Department of Clinical Microbiology at Hvidovre Hospital, Copenhagen, with community-acquired *E. coli* UTIs were identified via the laboratory management system. For each patient infected with an *E. coli* strain resistant to third-generation cephalosporins (3GC), we recruited one or two randomly chosen patients infected with a 3GC-susceptible strain. Patients above 85 years were excluded, since most of them lived in nursing homes, which are a healthcare rather than a community setting. Upon consent from the patient’s general practitioner, each patient (or a parent for patients below 18 years) was asked by telephone whether a dog or a cat lived in the household. In the case of a positive response, the patient was invited to participate in the study, and those agreeing to participate were provided with sampling material and instructed to dip a cotton swab into a fresh fecal deposit from their dog. Cat feces < 8 h old were sampled in litter trays using the same method. On the day of collection, swabs were shipped to the laboratory in commercial transport medium (BBL Cultureswab, Becton Dickinson, Franklin Lakes, NJ, USA). All samples were collected within two weeks after UTI diagnosis and processed in the laboratory within 48 h after collection.

### 4.2. E. coli Isolation, Identification, and Typing

Fecal swabs from pets were processed using antimicrobial selective methods to enhance detection of the *E. coli* strain causing human UTIs. If the owner was infected with a 3GC-resistant strain, samples were enriched overnight at 37 °C in MacConkey broth with 1 µg/mL cefotaxime and sub-cultured on MacConkey agar containing the same cefotaxime concentration. If the owner was infected with a 3GC-susceptible strain, a previously described selective direct plating method was used to detect *E. coli* displaying the same resistance profile as the UTI-causing strain [27]. In brief, discs containing the antibiotics to which the UTI strain was resistant (Oxoid, Basingstoke, UK) were applied to the surface of MacConkey agar streaked with the fecal swab. Upon overnight incubation at 37 °C, presumptive *E. coli* colonies growing in proximity of the discs or within the disc inhibition zone were cultured and identified by Maldi-TOF MS (Vitek MS RUO; bioMérieux, Marcy-l’Étoile, France). If colonies with different appearance were visible, we selected one for each colony type. All confirmed *E. coli* isolates were stored at −80 °C prior to further characterization.

Antibiotic susceptibility was tested by broth microdilution (Sensititre COMPAN1F plates, Thermo Fisher Scientific, Waltham, MA, USA) according to the Clinical and Laboratory Standards Institute (CLSI) [28]. Pet isolates were compared genetically to the UTI-causing strains by PFGE using XbaI (New England BioLabs, Ipswich, MA, USA) [29,30]. If 3GC-resistant isolates were yielded from both a patient and his/her pet, they were further characterized by PCR and sequencing for the presence of genes encoding ESBL (*bla*_TEM_, *bla*_SHV_, *bla*_CTX-M_) or plasmid-mediated AmpC (*bla*_CMY-2_) using previously described primers and protocols [31,32]. Plasmids harboring ESBL or AmpC genes were transformed into Genehog *E. coli*, followed by PCR-based replicon typing (PBRT) using a commercial kit (Diatheva, Cartoceto, Italy) and S1 nuclease PFGE for size determination [33].

### 4.3. Follow-Up Quantitative Microbiology Study

Patients infected with an *E. coli* strain that was indistinguishable by PFGE from a strain isolated from the feces of their pet were invited to participate in a follow-up study approximately 10 months after the initial UTI. They were instructed on how to collect feces from themselves and from their pet. All samples arrived at the laboratory the day after collection. Occurrence of the UTI-causing strain was quantitatively assessed by standard bacterial counts on MacConkey agar supplemented with antibiotics corresponding to the resistance profile of the strain. Antibiotic concentrations were selected to prevent growth of wild-type *E. coli* (8 µg/mL for ampicillin, 32 µg/mL for trimethoprim and 256 µg/mL for sulfadiazine). Up to six *E. coli* colonies from each sample were stored at −80 °C. Total coliforms (lactose-positive colonies) were counted on MacConkey agar without antibiotic, and a weighted average count of total and resistant coliforms was calculated.

### 4.4. Whole-Genome Sequencing and Genome Analysis

Human and pet isolates collected during the follow-up study were subjected to WGS. DNA was extracted using a MasterPure DNA Purification Kit (Epicentre, Madison, WI, USA). DNA libraries were prepared using the Nextera XT kit (Illumina Inc., San Diego, CA, USA) and sequenced on the V3 (2 × 300 bp) flow cell on the Illumina MiSeq platform to produce paired-end reads. Raw sequencing reads were assembled using SPAdes v.3.15.3 [34], and assemblies were annotated using Prokka v.1.14.6 [35]. The resulting .gff files were used to determine a core genome alignment using Roary [36]. Phylogenetic reconstruction was performed using IQ-TREE v.1.5.5 [37] with the best model found by the implemented ModelFinder and bootstrap analysis using 100 replicates. The tree was mid-rooted and visualized using iTOL v.6.7.4 [38]. Pairwise SNP distance among isolates was estimated using snp-dists (https://github.com/tseemann/snp-dists (accessed on 27 April 2023)). Assembled genomes were screened using ABRicate v1.0.1 (https://github.com/tseemann/abricate (accessed on 27 April 2023)) against the ResFinder [39], VirulenceFinder [40] and PlasmidFinder [41] databases, and alignment results with identity scores greater than 95% were selected as positive matches. MLST and phylogroup assignation was performed using mlst (https://github.com/tseemann/mlst (accessed on 27 April 2023)) and ClermonTyping [42], respectively.

### 4.5. Data Availability

WGS data for the project has been deposited to the NCBI Sequence Nucleotide Archive (SRA) under BioProject PRJNA376817.

## 5. Conclusions

Approximately 1 in 10 UTI patients living together with pets shared the strain causing infection with their pet, and in 1 case the strain persisted in the animal’s—but not the patient’s—feces for at least 10 months with minimal genomic rearrangements. Although this suggests zoonotic transmission, we cannot rule out the possibility of a reverse zoonosis (human to animal) or that the patient and pet acquired the shared *E. coli* strain from a common source. Nevertheless, these results have implications for primary care physicians, who, as part of their anamnestic workup, should enquire with recurrent UTI patients about dog ownership and provide guidance on hygiene precautions to prevent zoonotic transmission. Preventive measures include thorough hand washing or sanitizing before eating and after petting the animal or disposing of pet excrement; regular washing of pet bedding, toys, and bowls; maintaining clean household floors and furniture; and avoiding intimate contact with pets via face licking or bed sharing.

## Figures and Tables

**Figure 1 antibiotics-12-01269-f001:**
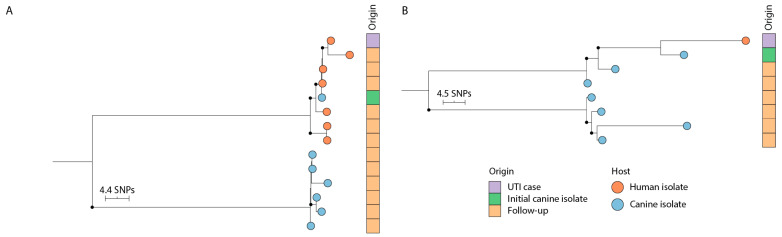
Midpoint-rooted phylogenetic tree based on core-genome phylogeny of strains from pair A (**A**) and pair B (**B**). Bootstrap values above 70% are illustrated by filled-in circles at the ends of branches. The scale bar represents the expected number of single nucleotide substitutions (SNPs).

**Figure 2 antibiotics-12-01269-f002:**
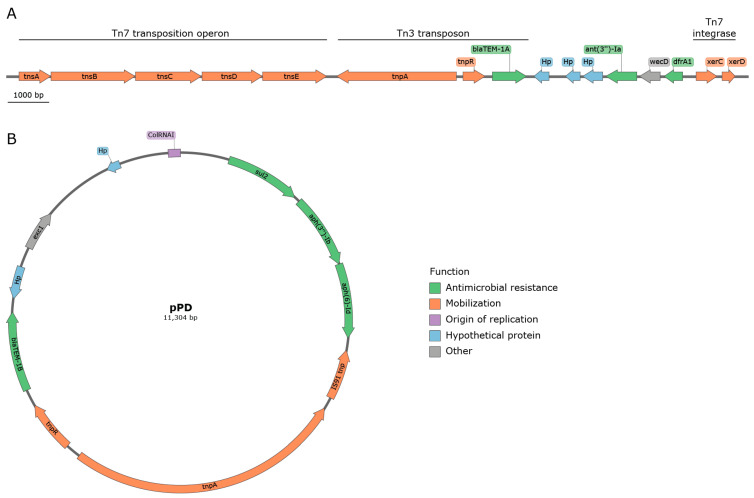
Genetic context of all antimicrobial resistance genes identified. In isolates of pair A (**A**), resistance genes are located within a Tn*7* transposon located on the chromosome. In isolates of pair B (**B**), resistance genes are located on a 11.3 kbp ColRNAI plasmid named pPD.

**Table 1 antibiotics-12-01269-t001:** Characterization of *E. coli* isolates from the seven patient/pet pairs for which antimicrobial selective culture indicated potential strain sharing.

Patient/Pet Pair	Host (Age/Gender) ^a^	Characterization of *E. coli* ^c^		
		Antimicrobial Resistance ^b^	ESBL	PFGE Type
A	H (69/♀)	AMP	−	1
	D (2/♂)	AMP	−	1
B	H (53/♀)	AMP, SXT	−	2
	D (8/♀)	AMP, SXT	−	2
C	H (66/♀)	(AMC), AMP, CFZ, CPD, (DOX), MAR, SXT	+	3
	D (12/♂)	AMP, CFZ, CPD, MAR, SXT	+	4
D	H (74/♀)	AMP, SXT	−	5
	D (11/♂)	None	−	6
E	H (74/♀)	(FOX), DOX, SXT	−	7
	C (5/♂)	None	−	8
F	H (67/♀)	(DOX), SXT	−	9
	D (1/♂)	SXT	−	10
G	H (44/♀)	(AMC), AMP, (DOX), GEN, SXT	−	11
	D (4/♀)	None	−	12

Abbreviations: AMC, amoxicillin/clavulanic acid; AMP, ampicillin; C, cat; CFZ, cefazolin; CPD, cefpodoxime; D, dog; DOX, doxycycline; GEN, gentamicin; H, human; MAR, marbofloxacin; SXT, trimethoprim/sulfamethoxazole, +, ESBL-positive; −, ESBL-negative. ^a^ Age is displayed in years. ^b^ Determined by broth microdilution. Brackets indicate isolates being intermediate to an antibiotic. ^c^ The human *E. coli* are clinical isolates from UTIs. The pet *E. coli* are commensal isolates from fecal samples that were collected within 2 weeks after the owner’s UTI episode.

**Table 2 antibiotics-12-01269-t002:** Origin and analysis of 22 genome-sequenced isolates obtained from patient/pet pairs A and B at the time of the patients’ UTI infection and approximately 10 months later.

Pair	Host	Sequenced Isolates (No.)	Month/Year of Isolation	Origin	ST	Phylogroup	ExPEC Virulence Genes	Resistance Genes
A	Patient	1	03/2014	UTI	ST998	B2	*papA*, *papC*, *sfaDSE kpsM* II	*ant(3″)-Ia*, *bla*_TEM-1A_, *dfrA1*
		6	01/2015	F				
	Dog	1	03/2014	F				
		6	01/2015	F				
B	Patient	1	03/2014	UTI	ST80	B2	*sfaD, kpsM* II	*aph(3″)-Ib, aph(6)-Id, bla*_TEM-1B_, *sul2*
	Dog	1	03/2014	F				
		6	02/2015	F				

Abbreviations: ExPEC, extra-intestinal pathogenic *E. coli*; F, feces; ST, sequence type; UTI, urinary tract infection.

## Data Availability

WGS data presented in this study are openly available in NCBI Sequence Nucleotide Archive (SRA) under BioProject PRJNA376817.

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
