# Peer review of "Dogs Can Be Reservoirs of Escherichia coli Strains Causing Urinary Tract Infection in Human Household Contacts"

_antibiotics, 2023, doi:10.3390/antibiotics12081269_

Round 1
Reviewer 1 Report
This is an interesting study.
Questions:
Authors should describe in discussion section limitations of the study:
Small number of pets that carry E. coli ( why?)
Low frequency of testing ( 10 month apart) instead monthly
Lack of f/u of patient's pets that were negative on the first screen
Reviewer 2 Report
Comments:
1. Only 19 pet owners were recruited in this study and finally persistence and evolution of UTI-causing strain in the two hosts were assessed by WGS analysis. The sample size is too small to prove that dogs are reservoirs of Escherichia coli strains causing human urinary tract infections.
2. Have the patients and dogs recruited in this study been treated with antibiotics within three months? The use of antibiotics can affect the resistance of E. coli and thus the reliability of the results of this study.
3. for the conclusion “This study shows that approximately one in ten UTI patients living together with pets shares the strain causing infection with the household pet and that the strain can persist in the animal feces for a long period (≥ 10 months) with minimal genomic rearrangements.” I am doubted about it.
4. As this study involved dogs and cats, it is unsuitable to use dogs can be reservoirs of…. It is suggested to use pets can be reservoirs of…
5. English writing should be polished. For example, the sentence “Two of the patients lived with two pets, leading to 21 pets sampled, which included six cats and 15 dogs” is confusing.
6. In line 189, there should be a blank space between “8” and “h”.
The English writing should be polished.
Reviewer 3 Report
1. Overall, the manuscript is a good one even though I wish that the sample size of 19 recruited was more.
2. Lines 41-50: Should this be part of the introduction or methodology?
3. Is there a chance of reverse zoonosis i.e. E. coli infection from humans (dog owners) to dogs? The authors need to clarify this.
Round 2
Reviewer 2 Report
The comments have not been addressed. The sample size is too small.
The English writing is good.